# Dermatopathological Challenges in Objectively Characterizing Immunotherapy Response in Mycosis Fungoides

**DOI:** 10.3390/dermatopathology12030022

**Published:** 2025-07-29

**Authors:** Amy Xiao, Arivarasan Karunamurthy, Oleg Akilov

**Affiliations:** Department of Dermatology, University of Pittsburgh Medical Center, Pittsburgh, PA 15213, USA

**Keywords:** mycosis fungoides, cutaneous T-cell lymphoma, biomarkers, tumor microenvironment, melanoma, treatment response quantification

## Abstract

In this review, we explore the complexities of objectively assessing the response to immunotherapy in mycosis fungoides (MF), a prevalent form of cutaneous T-cell lymphoma. The core challenge lies in distinguishing between reactive and malignant lymphocytes amidst treatment, particularly given the absence of uniform pathological biomarkers for MF. We highlight the vital role of emerging histological technologies, such as multispectral imaging and spatial transcriptomics, in offering a more profound insight into the tumor microenvironment (TME) and its dynamic response to immunomodulatory therapies. Drawing on parallels with melanoma—another immunogenic skin cancer—our review suggests that methodologies and insights from melanoma could be instrumental in refining the approach to MF. We specifically focus on the prognostic implications of various TME cell types, including CD8+ tumor-infiltrating lymphocytes, natural killer (NK) cells, and histiocytes, in predicting therapy responses. The review culminates in a discussion about adapting and evolving treatment response quantification strategies from melanoma research to the distinct context of MF, advocating for the implementation of novel techniques like high-throughput T-cell receptor gene rearrangement analysis. This exploration underscores the urgent need for continued innovation and standardization in evaluating responses to immunotherapies in MF, a field rapidly evolving with new therapeutic strategies.

## 1. Introduction

MF, the most common cutaneous T-cell lymphoma, is a challenging disease to monitor for treatment response by methods of traditional histology and immunohistochemistry. MF is characterized by patches, plaques, and tumors composed of skin-infiltrating lymphocytes. It is often difficult to assess differences in clinical and pathological features when distinguishing patch from plaque stage disease, and this differentiation is further muddied by variations in how plaques are defined clinically versus histopathologically [1]. Treatment of MF differs for early and advanced-stage disease. The early-stage disease is defined by patches or plaques in the skin with no blood, lymph node, or internal organ involvement, and is treated with skin-directed therapies. Advanced stage MF ranges from tumors in the skin only, to lymphoma that has spread to the blood, lymph nodes, and internal organs. Treatment options for advanced MF have most recently been heading in the direction of targeted immune therapy, like monoclonal antibodies and a newer generation of immune checkpoint inhibitors, toll-like receptor agonists, and chimeric antigen receptor T-cells.

One of the main challenges in monitoring treatment response in MF histologically is the identification and separation of reactive and malignant lymphocytes [2]. Currently, diagnosing MF with biopsy is the most reliable method. Neoplastic lymphocytes can be seen going into the epidermis (epidermotrophism), tagging along the basal layer, but may be present in the dermis as well, especially when MF progresses. They are typically small to medium in size, sometimes with an enlarged nucleus, and usually exhibit the T helper memory phenotype. The Ki67 index is usually low and roughly represents the number of neoplastic cells. The malignant lymphocytes can lose CD5 and CD7 markers, but there is no consistency in the expression of maturation markers between patients, and even within the same patient, the loss of maturation markers may occur as the disease progresses. Moreover, MF simply does not have pathological biomarkers of malignancy, when the diagnosis is rendered by exclusion based on the absence of normal phenotypical markers rather on the presence of the atypical protein. As the disease progresses, the malignant lymphocytes cluster in larger groups, express higher Ki67, and sometimes gain CD30 expression. Reactive lymphocytes are mainly found in the dermis, while malignant lymphocytes are epidermotropic; however, they become difficult to distinguish from each other when normal lymphocytes are engaged by immunotherapy (IMT) and become capable of infiltrating the epidermis where the malignant ones are. This histopathologic ambiguity is further amplified under immunotherapy, complicating the differentiation of reactive and malignant lymphocytes and leading directly to the next critical challenge: accurately evaluating residual disease and quantifying treatment response.

Another problem in monitoring treatment response in MF is the quantification of the response to IMT. One of the biggest challenges in quantifying response to IMT in MF is the evaluation of residual lymphoma cells. As illustrated in Figure 1, brentuximab vedotin (anti-CD30 antibody-drug conjugate) treatment may lead to apparent histologic response characterized by a loss of CD30 expression, even though neoplastic lymphocytes remain present. This antigen downregulation complicates post-treatment biopsy interpretation, as residual disease may be missed when relying solely on immunohistochemical markers.

While there is a general difficulty in differentiation between reactive lymphocytes and malignant cells, no assessment of the functional status of lymphoma cells (e.g., apoptotic or necrotic changes after IMT) is currently in use clinically. In addition, the presence of particular cell types in the tumor microenvironment (TME) can be predictive of response to therapy. For example, the presence of CD8 cells, NK cells, histiocytes, and plasmacytoid dendritic cells has been associated with a favorable response to therapy, while the presence of cancer-associated fibroblasts, B cells, and tertiary lymphoid structures has been associated with a poorer response in other cancers. There is very limited and unsystematized data available for MF so far.

Fortunately, newer histological techniques such as multispectral imaging and spatial transcriptomics are showing promise in solving some of these problems. Multispectral imaging allows for the identification and spatial characterization of multiple markers within a single tissue section, while spatial transcriptomics can be used to map the expression of genes within the context of the tissue architecture. Additionally, circulating tumor DNA (ctDNA) analysis via liquid biopsy and miRNA profiling are gaining traction in melanoma and show potential for translation into CTCL. These techniques may allow for a more comprehensive understanding of the TME and its response to therapy and may ultimately lead to a more accurate quantification of treatment response in MF. However, there is still much work to be performed in developing and validating these techniques, and in understanding the complex interplay between the TME and response to therapy in MF.

## 2. Utilizing Melanoma as a Model for Assessing the Effectiveness of IMT in MF

Much research is performed on how the TME in melanoma changes in response to IMT. Both MF and melanoma are immunogenic cancers that are localized to the skin and can be biopsied using similar methods [3,4]. The checkpoint inhibitor pathways like CTLA-4 and PD-1/PD-L1/PD-L2 are important in both MF and melanoma, but have been more extensively studied in melanoma due to their earlier utilization in IMT [5]. Pembrolizumab, one such inhibitor of this pathway, is used in both melanoma and MF. Moreover, standardized biopsy assessment frameworks developed in melanoma may serve as a useful model for evaluating immunotherapy effectiveness in MF and guiding clinical decision-making [6].

The TME in MF and melanoma share several similarities. In MF, the TME consists of fibroblasts, dendritic cells, macrophages, and T-cells that modulate tumor initiation, growth, and metastasis [7]. Recent studies have shown that the expansion of exhausted immune cells around malignant cells is associated with the progression of MF and inhibits immune activation and tumor clearance [8]. IMT has been shown to decrease the number of exhausted T-cells and increase the population of NK and CD8+ T-cells in MF [7]. However, as demonstrated in the recent EORTC trial of Atezolizumab in CTCL, baseline PD-L1 expression did not correlate with response to therapy, suggesting that commonly used biomarkers in melanoma may not reliably predict outcomes in MF [9]. The TME in MF appears to be an active participant in tumor progression and response to therapy, like melanoma, where the correlation between TME and treatment outcomes is well established. Despite limited research on the TME in MF, certain mechanistic parallels with melanoma—such as immune exhaustion and stromal contributions—may still help guide the development of more effective therapies, though direct translation of predictive biomarkers remains challenging.

While drawing parallels with melanoma is informative, it is also critical to define the clinical and research contexts in which MF biopsies are performed. In melanoma, biopsies may be taken to assess candidacy for immunotherapy, characterize immune phenotypes, or evaluate histologic response to treatment—particularly in research settings [10]. In MF, biopsies are more commonly used to confirm diagnosis or track progression rather than to guide immunotherapy decisions. However, as immune checkpoint inhibitors (ICIs) gain traction in MF, there is an increasing rationale for integrating baseline immune profiling into pre-treatment assessment frameworks. This may include evaluating CD8+ T-cell infiltration and PD-1 expression on TILs—both of which have been associated with response to ICIs in melanoma [11]. A recent review has emphasized the role of spatial immune biomarkers in melanoma and their potential relevance in other cutaneous malignancies [12].

Still, caution is warranted when applying melanoma-derived frameworks to MF. Unlike melanoma, where tumor and immune compartments are distinct, the malignant cells in MF are themselves immune cells—primarily CD4+ memory T-cells—blurring the line between immune activation and tumor persistence. This unique biology introduces added complexity in interpreting biopsy findings and may contribute to the limited efficacy of certain immunotherapies. For example, the modest results of CD47-directed therapy in MF likely reflect this underlying immunologic duality. Therefore, while melanoma can offer a structural model, developing MF-specific criteria that account for its immunologic nuances is essential to improving treatment evaluation and response prediction.

## 3. Prognostic Markers in the TME of Melanoma, and Implications for MF

In melanoma, the biopsies are assessed routinely for the density of various cell types in the TME, like CD8+ tumor-infiltrating lymphocytes, NK cells, B cells, histiocytes, and fibroblasts, to monitor their changes in subsequent biopsies for tumor progression or response to therapy [13]. This approach has been correlated with increased patient survival outcomes in melanoma and may have implications in monitoring response to IMT in other cancers, including MF. Our analysis of the literature highlights relevant findings regarding the prognostic implications of TME cells in other solid and hematologic tumors as well, emphasizing the utility of those findings for monitoring the response to IMT in MF (Table 1).

## 4. CD8+ Tumor-Associated Lymphocytes

In various tumor types, a high quantity of CD8+ TILs has been linked to better clinical outcomes [8]. In melanoma, the presence of CD8+ TILs was significantly associated with improved overall survival, and CD8+ TIL density at the invasive margin of melanoma metastases predicts a good response to therapy [14,15,16]. As a result, CD8+ TIL scoring is utilized in melanoma to quantify CD8+ TILs in the TME via standardized systems, such as those established by the Melanoma Institute of Australia (MIA; a score-based system on the density of CD8+ T-cells in the tumor center and invasive margin, as well as their distribution throughout the tumor) and Clark and colleagues (Clark system distinguishing three distinct CD8+ TILs patterns as absent, non-brisk, and brisk) [17]. Both systems have consistently demonstrated that increased CD8+ TIL levels are linked to better prognosis [17]. Increased CD8+ TIL infiltration, particularly when co-expressing PD-1, has been shown to correlate with improved response to anti-PD-1/PD-L1 therapies in melanoma [11,18,19], underscoring their value as potential biomarkers for treatment stratification. In addition, CD8+ TILs in MF also correlate with an increased survival rate [20,21] and are observed in the immediate vicinity of malignant cells following IMT (Figure 2).

Importantly, the findings from melanoma research highlight the significance of CD8+ TILs in the TME, which may have crucial implications for understanding and monitoring the response to IMT in MF.

## 5. NK Cells

NK cells play a crucial role in the activation of CD8+ T-cells through the secretion of IFN-gamma. Moreover, NK cells secrete several other pro-inflammatory cytokines and chemokines that enhance the response of the immune system against tumors, indicating their vital role in antitumor attack [22].

Intratumoral NK cell frequency in melanoma patients is associated with response to anti-PD-1 treatment and improved overall survival, highlighting the ability of NK cells to amplify the antitumor response [23]. Infiltration of NK cells in solid tumors is considered an indicator of good prognosis [24]. Similarly, the expansion of cytotoxic CD62L + NKG2A-CD107a + IFN-γ- NK cells after anti-CD47 immunotherapy was linked to a favorable clinical outcome in patients with tumor MF [25,26].

Given that both melanoma and MF present with accessible cutaneous lesions, they offer unique opportunities for the use of topically administered immunotherapies. TLR7/8 agonists, such as imiquimod and resiquimod, have been explored in both diseases, with promising local and systemic immunomodulatory effects [27]. In MF, topical resiquimod has been associated with increased activation of NK and CD8+ T-cells within treated lesions, alongside histologic regression and, notably, regression of distant, untreated lesions—suggestive of an abscopal effect. Similar findings have been reported in melanoma, where localized immune stimulation has occasionally triggered broader antitumor responses. These observations highlight the potential of topical immunotherapy not only as a localized treatment but also as a mechanism to systemically prime antitumor immunity, particularly relevant in early-stage or oligolesional disease.

## 6. Histiocytes and Tumor-Associated Macrophages

Traditionally, the presence of high levels of tumor-associated macrophages (TAMs) has been associated with a poor prognosis in cancer, but the development of anti-PD-1 therapy has changed this perspective [28,29]. Immunotherapy targeting PD-1 can activate senescent macrophages, transforming them into M2 phenotype and activating them against tumors. Therefore, when evaluating the prognostic value of TAMs, it is crucial to consider the type of treatment given to subjects. Recent studies have shown conflicting results on the role of TAMs as an indicator of poor prognosis in follicular lymphoma after incorporating rituximab, an anti-PD-1 agent, into treatment regimens [30]. Similarly, the presence of histiocytes at the advancing borders of melanoma has been linked to longer progression-free survival after anti-PD-1 therapy, indicating the activation and invasion of tumor margins and the picking up of necrotic debris [31]. In head and neck cancer, an increased proportion of tumor necrosis and giant cells/histiocytes in the tumor bed signals a better treatment response [32]. We have also observed dense histiocytic infiltration of a regional lymph node after successful anti-CCR4 treatment (Figure 3).

Rather than assessing individual populations of TAMs, the balance of M2/M1 TAMs in the TME seems more crucial. Therefore, the evaluation of the CD163/CD68 ratio on IHC can be used to indicate M2 polarization and worse disease outcomes in MF [33]. Since the subcutaneous administration of IFN-α2a was shown to modulate the immunomodulatory function of M2 TAMs, the assessment of the number of CXCL11-producing cells in the lesional skin of patients with advanced MF may be beneficial in predicting the efficacy of IMT [34].

While this review focuses primarily on MF, it is important to note that Sézary syndrome (SS), the leukemic variant of CTCL, presents distinct immunological dynamics in response to immunotherapy. Treatment with an anti-CCR4 monoclonal antibody (mogamulizumab) has been shown to induce a characteristic immunologic signature in more than 25% of patients with MF and SS, including increased expression of CD7, CD8, and CD163+ M2-polarized macrophages [35,36]. This immune activation profile is often associated with the development of mogamulizumab-associated rash (MAR), which has been correlated with improved treatment response. Notably, some patients demonstrate clinical benefit from immunotherapy even in the absence of MAR, underscoring the need to evaluate both histopathologic and systemic immune changes when assessing treatment efficacy in SS.

## 7. Plasmacytoid Dendritic Cells

Given the dual role of plasmacytoid dendritic cells (pDCs) in immune tolerance and induction of antitumor immunity [37], it is important to monitor their behavior and abundance in the context of mycosis fungoides (MF) following immunotherapy. Previous studies have demonstrated that pDCs are irregularly distributed throughout the dermal infiltrate in every stage of MF and are preferentially located close to the basement membrane. While there are more immature pDCs than mature ones, their prognostic value remains unknown. However, studies in other types of cancer have shown that recruitment of pDCs to the tumor microenvironment can lead to immune tolerance and poor survival [38,39,40,41,42]. For example, an increased number of pDCs in melanoma primary tumors and tumor-draining lymph nodes was associated with early relapse likely because they support Th2-driven inflammation, which creates a cytokine profile that is favorable for melanoma progression [43,44]. Therefore, tracing the abundance and activity level of pDCs after immunotherapy in MF could provide valuable insights into their role in the antitumor immune response and potential therapeutic targets.

## 8. Cancer-Associated Fibroblasts

Cancer-associated fibroblasts (CAFs) are another important cell type present in the tumor microenvironment that should be monitored after immunotherapy in MF. In several types of cancer, increased numbers of CAFs are associated with poorer prognosis [45,46,47,48], as they are responsible for modifying the extracellular matrix, making it stiffer and impeding T-cell infiltration [49], which leads to cancer progression and drug resistance. Similarly to TAMs, the prognostic relevance of CAFs in MF can depend on whether patients were treated or not. In treatment-naïve MF, CAFs were found to promote tumor cell migration and drug resistance [50], while post-treatment, the infiltration of the tumor bed by fibroblasts might be a consequence of tumor necrosis, resulting in neovascularization and fibrosis [13]. Monitoring the behavior of CAFs in MF after immunotherapy can provide valuable information on the effectiveness of the treatment and help predict patient prognosis.

## 9. B Cells

B cells have emerged as a complex component of the tumor microenvironment in MF, with stage-specific and context-dependent implications. While tumor-associated B cells (TABs) have shown positive prognostic value in some solid tumors such as melanoma, this does not directly translate to MF [51]. In plaque and tumor-stage MF, increased intratumoral B cell density has been correlated with poor prognosis and disease progression, particularly in cases of large-cell transformation [52]. Recent studies have shown that B cells may contribute to a tumor-supportive microenvironment by fostering malignant Th2 polarization and immune evasion [53,54]. These findings underscore a likely pathogenic role for B cells in advanced MF. In contrast, erythrodermic MF and Sézary syndrome are typically characterized by sparse B cell infiltrates in the skin and peripheral blood, and current data do not support a meaningful role for B cells in these subtypes. Moreover, regulatory B cells appear diminished in peripheral blood in advanced MF, suggesting systemic B cell dysfunction rather than infiltration [55]. The prognostic and functional role of B cells in MF, therefore, appears to vary by disease stage and subtype, and more stratified studies are needed to fully elucidate their contribution to immunotherapy response and disease course.

## 10. Tertiary Lymphoid Structures

Tertiary lymphoid structures (TLSs) play an important role in shaping the immune response to cancer [5]. TLSs are formed by B cells and other immune cells in response to chronic inflammation or infection, and they act as local sites of antigen presentation, T-cell activation, and antibody production. In melanoma, colorectal cancer, and breast cancer [56,57,58], TLSs have been associated with increased survival, and in melanoma, they have been shown to be a positive prognostic factor in response to immune checkpoint blocking therapy [6]. In PD-1-responsive CTCL patients, there appears to be a trend towards greater lymphatic-enriched stroma, which may indicate the presence of TLSs [59].

## 11. Novel Methods for Monitoring the Response to IMT

New methods for monitoring the response to IMT are continuously being developed. One such method is multispectral imaging (MSI), which has been shown to be a powerful and reliable technique for screening several antigens on a single tissue section and analyzing the immune infiltrate in conjunction with immune checkpoint molecules. MSI has been demonstrated to be strongly predictive of the response to immunotherapy in melanoma [60], and it may prove to be useful in monitoring the response to immunotherapy in MF as well. Multispectral immunohistochemistry has demonstrated that the ratios of CD8+ to FOXP3 and CD8+ to programmed death-ligand 1 (PD-L1) were significantly predictive of response to immunotherapy in melanoma, suggesting that this technique could be useful in MF as well [61]. Importantly, the Stanford group recently published some of the most significant findings to date regarding immune cell topography as a predictor of response to PD-1 therapy, emphasizing that spatial organization—rather than simple expression levels—may better forecast therapeutic outcomes [59]. In parallel, a phase II study of atezolizumab (a humanized IgG antibody that binds PD-L1) in CTCL found no correlation between PD-L1 expression and treatment response, further supporting the need for more sophisticated spatial and cellular analyses, such as MSI, over single-marker assessments [62].

Another method that can be used to monitor response to immunotherapy in MF is high-throughput T-cell receptor gene rearrangement analysis. This technique involves sequencing the TCR genes in the biopsy specimen to determine the clonality of the T-cells present, and it has been shown to be highly specific in the assessment of both clonality and T-cell fractions in skin biopsies and can show how many T-cells are tumor vs. reactive by comparison of monoclonal and polyclonal peaks, respectively [63]. A tumor clone frequency (TCF) above ~25% in early-stage MF was demonstrated to be an independent predictor of disease progression and poorer survival [64]. Further, TCR clonality analysis showed that pronounced copy number variations alongside TCR rearrangements in early-stage MF also carry prognostic significance [65].

However, there are limitations to this technique. For example, CR clonality can also be detected in benign conditions such as reactive monoclonal lymphocytosis, and malignant T-cells in MF are not necessarily purely monoclonal—especially as the disease progresses and clonal heterogeneity increases [66]. Moreover, a standardized cutoff value for minimal residual disease based on TCR gene rearrangement has yet to be established. Nonetheless, high-throughput sequencing technologies have significantly enhanced the diagnostic and prognostic utility of this approach. For example, de Masson et al. demonstrated that a tumor clone frequency above approximately 25% in early-stage MF was an independent predictor of disease progression and poorer survival [64]. More recently, Cieslak et al. applied a nanopore-based sequencing method and revealed that TCR gene rearrangements in early MF are frequently associated with pronounced copy number variations (CNVs), which may serve as an additional prognostic factor [65]. Finally, combining clonality data with high-resolution cell phenotyping—such as single-cell RNA sequencing—can further refine our understanding of early disease pathogenesis. Notably, large-plaque parapsoriasis lesions (considered part of the MF spectrum) often harbor dominant T-cell clones and distinct stromal–immune microenvironments, suggesting actionable early biomarkers [67].

In melanoma, a structured histologic framework has been developed to stratify responses to immunotherapy, considering not only tumor regression but also changes in the tumor microenvironment. Notable features include increased tumor-infiltrating CD8+ T-cells, tumor necrosis, fibrosis, melanophage accumulation, and spatial organization of immune infiltrates [18,19]. Models such as the Clark system and the Melanoma Institute of Australia (MIA) criteria incorporate the density and distribution of immune cells in both the tumor center and invasive margin. These histologic and spatial metrics are now used to monitor therapy response in clinical trials. Applying a similar multiparametric framework in MF could enhance response stratification, particularly as immunotherapy becomes more widely used. In CTCL, emerging data have shown correlations between spatial arrangements of PD-1+ CD4+ T-cells, tumor cells, and Tregs and response to pembrolizumab [59]. A rapid decline in exhausted BTLA+ CD4+ bystander T-cells was associated with positive response to anti-CD47 therapy in MF [7]. Moreover, increased proximity of CD4+ T-cells to tumor cells and greater distance of Tregs from malignant cells have been linked to better effector function. These observations suggest that integrating immune spatial metrics—modeled after melanoma—with emerging MF-specific features may help create a practical histopathologic response assessment system tailored to MF.

In addition to tissue-based techniques, emerging technologies such as circulating tumor DNA (ctDNA) analysis via liquid biopsy and miRNA profiling are gaining traction in melanoma and offer potential avenues for translational application in MF [68]. Liquid biopsies provide a minimally invasive approach to track dynamic tumor evolution and immune resistance mechanisms, while miRNA signatures may help identify predictive or prognostic biomarkers. These tools could be especially valuable in settings where repeated skin biopsies are impractical. However, in MF—where disease is primarily confined to the skin and directly visible—clinical assessment remains central, and there is currently no consensus or guideline recommending systemic intervention in the absence of visible lesions. Moreover, as MF treatments are largely non-curative and focused on disease control, the utility of detecting minimal residual disease through liquid biopsy or miRNA profiling remains unclear at this stage. Nonetheless, integrating such techniques with spatial immune profiling could enrich our understanding of the tumor microenvironment and may have future relevance as therapeutic strategies evolve.

In conclusion, the response to immune checkpoint inhibitor therapy in MF is a complex process that involves the interplay between tumor cells, immune cells, and the tumor microenvironment. New methods for monitoring the response to immunotherapy are continuously being developed, including MSI and high-throughput TCR gene rearrangement analysis, liquid biopsy, and mRNA profiling. In addition, a framework for stratifying the response to immunotherapy that takes a broader picture of the tumor microenvironment into account exists in melanoma and may be useful in MF as well. A combination of these approaches may be the most effective way to monitor the response to immunotherapy in MF and improve patient outcomes.

## 12. Conclusions

Assessing treatment response in MF remains underdeveloped, particularly as immunotherapies are increasingly incorporated into clinical care. Unlike melanoma—where standardized frameworks exist for evaluating the TME, spatial immune organization, and biomarker expression—MF lacks comparable tools. This is not due to a lack of relevance, but rather to the unique biological complexity of MF, where the malignant cells are themselves immune cells, blurring the distinction between tumor and immune response.

The purpose of this review is not to draw direct comparisons between MF and melanoma, but to underscore how much progress has been made in melanoma and to explore which of those advances might be thoughtfully adapted to MF. Techniques such as dual staining, spatial profiling, multispectral imaging, and high-throughput TCR rearrangement analysis—widely used or validated in melanoma—hold promise for MF if tailored to its biology. These tools can help overcome the current limitations of relying solely on clinical observation and traditional histopathology, particularly when immunotherapy alters antigen expression without eliminating disease.

In summary, MF is significantly behind other cutaneous malignancies in developing objective methods for assessing treatment response. By learning from melanoma and adapting its successful methodologies to the unique context of MF, we can begin to build a much-needed framework for a response evaluation that reflects the complexity of this disease and supports better therapeutic decisions.

## Figures and Tables

**Figure 1 dermatopathology-12-00022-f001:**
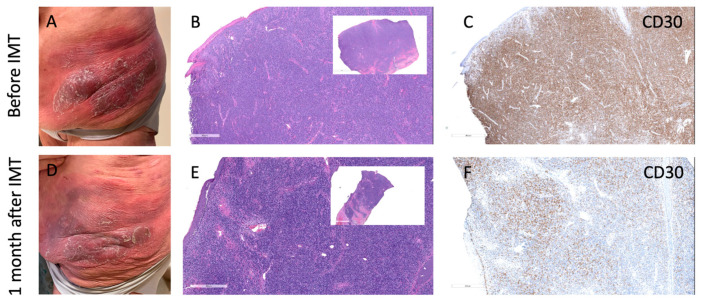
Antigen loss despite persistent infiltrate following brentuximab vedotin treatment in MF. (**A**–**C**) Clinical photo (**A**), H&E (**B**), and CD30 immunohistochemistry (**C**) before immunotherapy (IMT) showing dense plaque with strong CD30 expression. (**D**–**F**) One month after BV therapy: clinical improvement (**D**), persistent dense lymphoid infiltrate on H&E (**E**), but markedly reduced CD30 expression (**F**), indicating loss of CD30 antigen while lymphoid infiltrate remains. This pattern illustrates the challenge of evaluating residual disease after targeted immunotherapy, as cell presence may persist despite antigen loss.

**Figure 2 dermatopathology-12-00022-f002:**
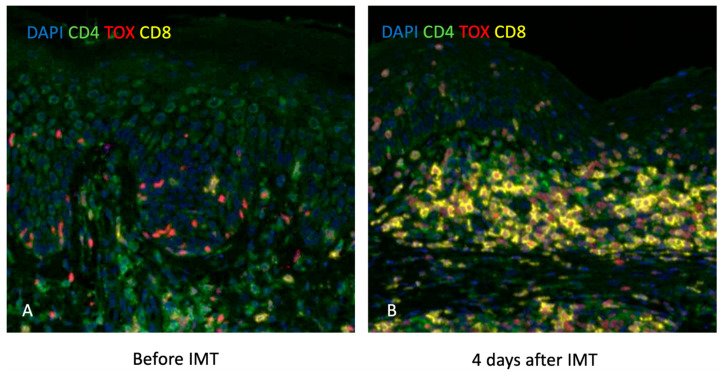
Slides with DAPI, CD4, TOX, and CD8 stains showing increased CD8+ T-cells in the immediate vicinity of malignant cells 4 days after IMT in a patient with MF. (**A**) Pre-treatment biopsy taken before initiation of IMT shows sparse CD8+ T-cell infiltration in the lesional skin, with scattered CD4+ and TOX+ cells. (**B**) Biopsy taken 4 days after IMT reveals a marked increase in CD8⁺ T-cell infiltration (yellow), particularly in the tumor-rich dermal areas, indicating early immune activation and potential anti-tumor response.

**Figure 3 dermatopathology-12-00022-f003:**
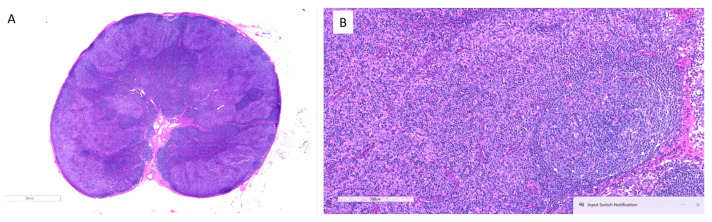
Dense histiocytic infiltration of a regional lymph node after successful treatment with anti-CCR4 in a patient with MF. (**A**) Low-power view of the entire lymph node section shows overall architectural effacement with diffuse infiltrates, indicating significant immune cell activity post-treatment. (**B**) Higher magnification of the same node reveals dense sheets of histiocytes replacing nodal parenchyma, consistent with a reactive histiocytic response following effective tumor clearance.

**Table 1 dermatopathology-12-00022-t001:** Prognostic roles of cell populations in melanoma and mycosis fungoides tumor microenvironments.

Cell Type/Structure	Melanoma Prognostic Role	MF Prognostic Role	Unmet Clinical/Research Needs in MF
CD8+ tumor-associated lymphocytes (TILs)	High density is associated with favorable prognosis and improved immunotherapy response.	Presence and spatial organization may predict outcome, but findings are inconsistent across stages.	Standardized methods to assess spatial distribution and functionality; correlation with response to immunotherapy.
Natural killer (NK) cells	Low frequency and impaired function correlate with worse outcomes	Can lyse malignant cells, but reduced cytotoxic activity may accompany disease progression.	Better characterization of NK subsets and their interaction with malignant T-cells in MF lesions.
Histiocytes/tumor-associated macrophages (TAMs)	M2-like polarization linked to poor prognosis and immune evasion.	TAMs exhibit M2-skewed, immunosuppressive phenotype contributing to progression.	Functional profiling of TAMs in early vs. advanced MF; role in immunotherapy resistance.
Plasmacytoid dendritic cells (pDCs)	Produce type I IFNs, may support both antitumor immunity and immune suppression.	Accumulate in MF lesions; implicated in immune suppression and poorer outcomes.	Need for mechanistic studies defining pDC function and impact on immunotherapy.
Cancer-associated fibroblasts (CAFs)	Promote fibrosis, tumor growth, and resistance to therapy.	Contribute to dermal fibrosis and immune modulation; associated with worse prognosis.	Lack of spatial/functional profiling in MF; role in creating immune-excluding niches.
B cells	Dual role: antibody production (anti-tumor) vs. regulatory B cells (pro-tumor); context-dependent.	High intratumoral B cell density correlates with poor prognosis in advanced MF and large-cell transformation.	Further stratification by MF stage; mechanisms of Th2 support and potential for targeting B cell interactions.
Tertiary lymphoid structures (TLSs)	Associated with improved immune infiltration and ICI response.	Rarely observed; emerging evidence suggests a role in antitumor immunity but not well-defined.	Larger studies needed to determine TLS presence, structure, and prognostic value in MF.

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
