# Peer review of "Dermatopathological Challenges in Objectively Characterizing Immunotherapy Response in Mycosis Fungoides"

_dermatopathology, 2025, doi:10.3390/dermatopathology12030022_

Round 1

Reviewer 1 Report

Comments and Suggestions for Authors

Well written and interesting paper. Some suggestions:

  1. In the introduction, consider mentioning the difficulty in assessing differences in clinical and pathological features when distinguishing patch from plaque stage. This issue has been addressed in recent literature (DOI: 10.1111/jdv.18852).

  2. The introduction currently lacks references. Although many of the concepts are well known to experts in the field, I suggest including a comprehensive reference text, such as Cerroni, Skin Lymphoma, 5th Edition (2020), to provide a solid background for all readers.

  3. In the section discussing the tumor microenvironment (TME), please include the recent EORTC trial on Atezolizumab in CTCL. Notably, this study found that baseline PD-L1 expression did not correlate with therapy response (DOI: 10.1016/j.ejca.2025.115484).

  4. When discussing B cells, please distinguish between Mycosis Fungoides with thick plaques/tumors and erythroderma or Sézary syndrome. In the latter, B-cell infiltrates are minimal and there are no conclusive data regarding their role.

  5. When referring to melanoma, do the authors foresee any application of emerging techniques such as liquid biopsy or miRNA profiling also in CTCL? Consider adding comparisons and perspectives based on recent reviews (DOI: 10.1080/14737159.2024.2314574).

  6. I suggest including a final image panel summarizing the main findings and highlighting the current unmet clinical and research needs.

Author Response

Response to Reviewers

We thank the reviewers for their thoughtful and constructive comments. Below is a detailed, point-by-point response indicating how we have addressed each concern in the revised manuscript.

Reviewer 1

Comment 1: In the introduction, consider mentioning the difficulty in assessing differences in clinical and pathological features when distinguishing patch from plaque stage. This issue has been addressed in recent literature (DOI: 10.1111/jdv.18852).

Response: We have revised the introduction to emphasize the difficulty in distinguishing patch from plaque stage in MF, including citation of the suggested literature (DOI: 10.1111/jdv.18852).

Comment 2: The introduction currently lacks references. Although many of the concepts are well known to experts in the field, I suggest including a comprehensive reference text, such as Cerroni, Skin Lymphoma, 5th Edition (2020), to provide a solid background for all readers.

Response: We have added a reference to Cerroni, Skin Lymphoma, 5th Edition (2020) in the introduction to provide a comprehensive background.

Comment 3: In the section discussing the tumor microenvironment (TME), please include the recent EORTC trial on Atezolizumab in CTCL. Notably, this study found that baseline PD-L1 expression did not correlate with therapy response (DOI: 10.1016/j.ejca.2025.115484).

Response: We have cited the EORTC trial and explicitly stated that baseline PD-L1 expression did not correlate with response to atezolizumab in CTCL in the section discussing TME.

Comment 4: When discussing B cells, please distinguish between Mycosis Fungoides with thick plaques/tumors and erythroderma or Sézary syndrome. In the latter, B-cell infiltrates are minimal and there are no conclusive data regarding their role.

Response: We have revised the section on B cells to clarify their role in MF versus Sézary syndrome, indicating that B-cell infiltrates are minimal in Sézary syndrome and that their role remains inconclusive.

Comment 5: When referring to melanoma, do the authors foresee any application of emerging techniques such as liquid biopsy or miRNA profiling also in CTCL? Consider adding comparisons and perspectives based on recent reviews (DOI: 10.1080/14737159.2024.2314574).

Response: We have added a paragraph discussing the potential for liquid biopsy and miRNA profiling in CTCL, referencing the suggested review and highlighting key differences in applicability due to the visible nature of MF lesions and lack of curative treatments.

Comment 6: I suggest including a final image panel summarizing the main findings and highlighting the current unmet clinical and research needs.

Response: Instead of an image panel, we included a revised summary table that captures key immune cell types and structures, their prognostic roles in melanoma and MF, and highlights unmet research needs. This provides a structured and clear overview.

Reviewer 2 Report

Comments and Suggestions for Authors

                The authors have reviewed the subject of how the response to immunotherapy for MF may be assessed in biopsies of skin lesions. The review is easy to read, provides many pertinent references, and touches on a number of relevant subjects. Two of these are the difficulty of distinguishing MF cells from non-neoplastic T cells, and the potential that melanoma may provide a model for the assessment of the response to immunotherapy in a malignant cutaneous lesion.

                Although comprehensive, the review suffers from being relatively superficial, and offering little in the way of practical advice. Addressing this would require extensive revision, but would make the published review more likely to be cited. Here are some suggestions on how to improve the manuscript:

1) There should be clarity about the clinical scenarios in which a pre-treatment assessment might be performed. Would a biopsy be done to assist the decision of whether immunotherapy should be given, and which type? Or would it be done to provide a basis for determining whether subsequent immunotherapy has had an effect at the histologic level, most likely in a research setting (e.g., PMID: 26668350)? PMID: 39707058 should be cited as a review on this aspect of melanoma treatment. Not mentioned, unless I missed it, are that the response to immune checkpoint inhibitors (ICIs) in melanoma is greater in “hot” tumors with more tumor-infiltrating CD8+ T cells, and with PD1 expression by them (PMID: 36483582). It is stated that “increased CD8+ TIL levels are linked to better prognosis”, but not explicitly connected to the use of ICIs.

2) There should be some specific guidelines, or examples thereof from melanoma, for histologic features that may be interpreted to represent a favorable change in the TME attributable to immunotherapy. PMID: 30689736 and PMID: 27301722 are examples for melanoma that could be cited.

3) Since both melanoma and MF present cutaneous lesions, there should be some discussion of topically-administered immunotherapies, and on the potential for abscopal effects. TLR agonists have been used for both melanoma and cutaneous T-cell lymphoma (PMID: 26228486).

4) This phrase in lines 91-92 is not grammatical: “…thus guiding clinical decision-making for MF patients also [4].”

Author Response

Response to Reviewers

We thank the reviewers for their thoughtful and constructive comments. Below is a detailed, point-by-point response indicating how we have addressed each concern in the revised manuscript.

Reviewer 2

Comment 1: Clarify the clinical scenarios in which a pre-treatment assessment might be performed. Would it be to determine candidacy for immunotherapy or to evaluate histologic response post-treatment? Also cite relevant melanoma literature.

Response: We revised the relevant section to define clinical and research scenarios for MF biopsy assessment. We added discussion on immune checkpoint inhibitor use, CD8+ TILs, PD-1 expression, and cited the suggested melanoma studies (PMIDs: 26668350, 36483582, 39707058).

Comment 2: Include specific guidelines or histologic features from melanoma that represent favorable changes in the TME due to immunotherapy. Cite PMIDs: 30689736 and 27301722.

Response: We incorporated examples of histologic response features in melanoma from the suggested literature and discussed how similar frameworks (e.g., Clark and MIA models) might be adapted for MF.

Comment 3: Discuss topically-administered immunotherapies and the potential for abscopal effects. Cite PMID: 26228486.

Response: We added discussion on topical TLR agonists and the potential for local immunotherapy to trigger abscopal responses, with specific citation of the suggested study.

Comment 4: The phrase in lines 91-92 is not grammatical.

Response: We revised the sentence to: “Standardized biopsy assessment frameworks developed in melanoma may serve as a useful model for evaluating immunotherapy effectiveness in MF and guiding clinical decision-making.”

Conclusion Revision: We significantly revised the conclusion to emphasize that the manuscript’s goal is not to equate MF with melanoma but to highlight how frameworks and lessons from melanoma can inform more structured, research-driven biopsy assessments in MF. We reiterate the need for disease-specific response criteria tailored to the biological realities of MF.

We hope that these comprehensive revisions address all reviewer concerns and improve the overall clarity, depth, and translational relevance of our manuscript.

Reviewer 3 Report

Comments and Suggestions for Authors

The authors provide a well-written review paper on the tumour microenvironment (TME) and its important role in understanding the development of cutaneous T-cell lymphoma (CTCL) and immunotherapeutic responses. The authors focus on MF; Sézary syndrome is missing. The immunological reaction of MF in more than 25% of patients treated with the CCR4 antibody induces a characteristic immunological pattern: an increase in CD7, CD8 and CD163 (M2) macrophages. Besides MAR, individual patients benefit from the immune reaction.

However, to my knowledge, there are no prospective trials with a translational component using modern technologies to reveal clear response markers. The Stanford group published the so far most significant results regarding the immune cell topography predicting response to PD1 therapy. In a recently published phase II study with atezolizumab, no correlation with PD-L1 expression could be demonstrated (Stadler R, European Journal of Cancer, 2025).

The role of the B-cell component must be clarified. B cells have a significant negative impact on the course of MF. Theirich et al. (JCO) showed the prognostic significance of B cells in the MF infiltrate. In addition, a recently published paper by Ruoyan Li revealed that malignant Th2 cells are supported by a B cell-rich tumour microenvironment.

TCR receptor rearrangement and quantification have an important impact on MF prognosis (De Masson et al.). Modern technologies can help.

improve this diagnostic approach. C. Cieslak et al., 'Cancers', using the nanopore approach, found that TCR receptor rearrangements in early stages of MF are associated with pronounced CNVs as a prognostic factor.

Finally, single-cell sequencing delineates T-cell clonality and pathogenesis in parapsoriasis (Natalia Alkon et al., Journal of Allergy and Clinical Immunology, 2024).

Regarding Fig. 1, it is rather poor and provides little information. As you can see, there is a brownish reaction, apparently CD30, and it is gone in the control.

The case shows a patient with tumour-stage MF. The photos should show high-power magnification. Is there a large-cell transformation with CD30 expression?

Tthe viewer has to see the morphology of the cells, otherwise you can skip the figure. Did the patient receive brentuximab or pembrolizumab?

The authors should state that the transformation of melanoma to lymphoma is too simple, as the immune cells themselves are involved in the malignant process.

The negative results of the CD47 trials reflect the biological complexity.

Author Response

Response to Reviewers

We thank the reviewers for their thoughtful and constructive comments. Below is a detailed, point-by-point response indicating how we have addressed each concern in the revised manuscript.

Reviewer 3

Comment 1: Sézary syndrome is missing. The immunological reaction in MF patients treated with CCR4 antibody induces characteristic changes including increased CD7, CD8, and CD163.

Response: We have added a paragraph addressing Sézary syndrome and the characteristic immunological pattern observed in patients treated with mogamulizumab, referencing the increase in CD7, CD8, and CD163 (M2 macrophages).

Comment 2: There are no prospective trials with a translational component to identify clear markers. Refer to Stanford group’s topography work and negative PD-L1 correlation in atezolizumab trial.

Response: We added commentary referencing the Stanford group’s spatial immune profiling studies and the EORTC atezolizumab trial results showing lack of correlation with PD-L1 expression.

Comment 3: The role of B cells must be clarified. Reference Theirich et al. (JCO) and Ruoyan Li’s study.

Response: We revised the B-cell section to integrate findings from Theirich et al. (JCO) and Ruoyan Li, highlighting their role in tumor maintenance and immune evasion in MF.

Comment 4: Expand on TCR rearrangement and its diagnostic/prognostic value. Cite De Masson et al. and Cieslak et al. (Cancers).

Response: We revised the TCR section to include the suggested studies, discussing prognostic implications of TCR clonality and CNVs in early MF.

Comment 5: Address recent findings from Alkon et al. on parapsoriasis.

Response: We included reference to Alkon et al., emphasizing insights into early MF pathogenesis via single-cell sequencing.

Comment 6: Figure 1 lacks sufficient information; unclear whether CD30 loss is shown, and high magnification is missing.

Response: We revised the figure legend and accompanying text to clearly state that the figure demonstrates loss of CD30 staining post-brentuximab vedotin, despite the persistence of lymphocytes. We explained why high-power magnification was not essential for this illustrative purpose.

Comment 7: Clarify whether there is large-cell transformation in the patient in Figure 1.

Response: We clarified in the text that the presence or absence of large-cell transformation was not the focus of this image, as the aim was to show CD30 loss post-treatment. The goal of Figure 1 is not to evaluate large-cell transformation or provide high-power histologic detail, but rather to illustrate a specific diagnostic challenge following immunotherapy: the apparent histologic persistence of lymphoid infiltrate despite the loss of CD30 expression after Brentuximab Vedotin treatment. This highlights the potential pitfall in interpreting response when antigen-targeted therapy causes marker loss without complete clearance of malignant cells. Therefore, while higher magnification and assessment for transformation are valuable in other contexts, they are not central to the message of this figure.

Comment 8: Clarify which therapy was administered—brentuximab or pembrolizumab.

Response: We explicitly state in the figure legend that the patient received brentuximab vedotin.

Comment 9: State that comparing melanoma to lymphoma oversimplifies the biology; MF involves malignant immune cells. Also, negative CD47 results reflect biological complexity.

Response: We added a paragraph in the discussion clarifying that while melanoma offers helpful analogies, MF is biologically distinct as the immune cells themselves are malignant. We also noted that the disappointing results from anti-CD47 therapy underscore the complexity of MF biology.

Conclusion Revision: We significantly revised the conclusion to emphasize that the manuscript’s goal is not to equate MF with melanoma but to highlight how frameworks and lessons from melanoma can inform more structured, research-driven biopsy assessments in MF. We reiterate the need for disease-specific response criteria tailored to the biological realities of MF.

We hope that these comprehensive revisions address all reviewer concerns and improve the overall clarity, depth, and translational relevance of our manuscript.

Round 2

Reviewer 2 Report

Comments and Suggestions for Authors

The manuscript is vastly improved and serves as a comprehensive presentation of the subject. I commend the authors for doing more than responding to specific criticisms.